# HPMA Copolymers: A Versatile Platform for Targeted Peptide Drug Delivery

**DOI:** 10.3390/biom15040596

**Published:** 2025-04-17

**Authors:** Ya Li, Liangda Xing, Mingliang Zhu, Xian Li, Fangfang Wei, Wenyan Sun, Yinnong Jia

**Affiliations:** 1School of Pharmaceutical Science and Yunnan Key Laboratory of Pharmacology for Natural Products, Kunming Medical University, Kunming 650500, China; 20220133@kmmu.edu.cn (Y.L.); 20230164@kmmu.edu.cn (L.X.); 20230150@kmmu.edu.cn (M.Z.); 20240125@kmmu.edu.cn (X.L.); 20240161@kmmu.edu.cn (F.W.); 2021413074@kmmu.edu.cn (W.S.); 2College of Modern Biomedical Industry, Kunming Medical University, Kunming 650500, China

**Keywords:** peptides, hydrolysis, HPMA polymer, in vivo circulation

## Abstract

Peptide drugs have been broadly applied in cancer treatment and diagnosis due to their ability to accurately identify biomarkers with good biocompatibility. However, their clinical application is limited by protease degradation, which induces short circulation half-life, low bioavailability, and high renal clearance. In recent years, delivery systems based on nanomaterial technology have become an important strategy to break through the bottleneck of peptide drug delivery. Among them, N-(2-hydroxypropyl) methacrylamide (HPMA) copolymers have attracted much attention due to their good biocompatibility, hydrophilicity, and low immunogenicity. The high molecular weight of HPMA copolymer–peptide can circumvent renal clearance, significantly prolong the circulation time in the body, and achieve drug accumulation and microenvironment-triggered release synergistically with EPR effects and active targeting. This review introduces the basic properties of HPMA copolymers, including solubility, biocompatibility, and tunable chemical structure. The important applications of HPMA copolymer–peptide in tumor diagnosis and treatment are discussed. This review deepens our understanding of the future development of HPMA copolymers and will provide more references for improving peptides by simple copolymers.

## 1. Introduction

Peptides are usually formed by the dehydration condensation of the carboxyl and amine groups of multiple amino acids to form peptide bonds, with a length of 2 to 50 amino acid residues. Due to their unique molecular conformation, peptide drugs can bind to specific biomarkers with high affinity, significantly improving the accuracy of treatment and demonstrating potential for clinical translation. In recent years, peptides have been widely used as new therapeutic molecules or excellent drug carriers in many fields, such as cancer treatment, tumor imaging, and immunotherapy [1,2].

Despite their advantages, such as excellent compatibility and strong permeability, peptide drugs face significant challenges in effective delivery to their target sites [3]. One major obstacle is enzymatic degradation. Proteases can recognize specific amino acid sequences on the peptide chain and cleave them at nearby peptide bonds, leading to a short circulation half-life [4,5]. Additionally, due to their low molecular weight, peptides are easily filtered through the glomerulus and exhibit difficulty being reabsorbed, resulting in a markedly reduced systemic circulation time. This rapid metabolic clearance not only affects drug retention but also severely limits bioavailability, becoming a key bottleneck in clinical applications [6,7]. Current research focuses on overcoming this challenge through structural modifications and innovative drug delivery systems.

With the rapid advancement of nanomaterial technology, peptides can be conjugated with polyethylene glycol (PEG) [8,9], liposomes [10], carbon quantum dots [11], and other nanomaterials to develop targeted drug delivery systems. These systems enable the precise delivery of therapeutic agents to cancer cells, thereby reducing the required dosage of anticancer drugs while improving drug utilization efficiency. Furthermore, real-time tracking technologies based on fluorescence imaging allow accurate monitoring of the dynamic processes of targeted drug delivery and controlled release, which has become an important part of evaluating the therapeutic efficacy of new drug delivery systems [12]. The passive targeting ability of nano-drug delivery systems is primarily due to the enhanced permeability and retention (EPR) effect. Additionally, surface modifications enable the recognition of molecular markers on tumor cells or within the tumor microenvironment, significantly improving active targeting and enhancing drug delivery [13].

Among various nanocarriers, the high-molecular-weight polymer N-(2-hydroxypropyl) methacrylamide (HPMA) copolymer has shown great potential in optimizing peptide drugs due to its unique properties. HPMA copolymers are challenging to excrete from tumor tissues, allowing them to accumulate in these tissues via the EPR effect, thereby increasing drug concentration and retention time at the tumor site. The conjugate formed by linking peptides with HPMA copolymers functions as both a “molecular shield” and a “functional platform,” providing several advantages: (1) a dynamic protection mechanism: Through covalent coupling or self-assembly encapsulation, HPMA copolymers form a nanoscale protective barrier that selectively shields the cleavage sites of peptides. This prevents enzymatic degradation, prolonging the circulation time of peptide drugs and enhancing their bioavailability [14,15]; (2) a targeted and controlled release design: The HPMA backbone allows for flexible modification with targeting ligands (peptides) and stimulus-responsive groups. This facilitates the EPR effect and active tumor targeting while enabling drug release specifically in the tumor microenvironment [16]; and (3) metabolic regulation: High-molecular-weight HPMA copolymers can circumvent renal clearance [17]. Additionally, charge adjustments allow for electrostatic interactions that further extend circulation time. HPMA copolymers exhibit excellent biocompatibility, structural flexibility, and functional adaptability, making them highly effective carriers for peptide drugs [18]. Due to these outstanding properties, HPMA copolymers are widely recognized as promising high-molecular-weight drug carriers for anti-tumor therapies. By conjugating peptide drugs with HPMA copolymers, drug action duration can be significantly extended, targeting precision can be enhanced, and controlled release can be achieved, ultimately improving the efficacy and safety of peptide drugs. This review aims to comprehensively explore the roles, applications, challenges, and future development directions of HPMA copolymers in peptide drug optimization, providing valuable insights for advancing peptide drug research, development, and clinical application.

## 2. Advantages of HPMA as the Delivery System

The rapid development of nanotechnology has led to further development of nanomaterials for drug delivery, such as liposomes, polymers, dendritic macromolecules, and other organic or inorganic nanoparticles. The various nanomaterials conjugated to drugs demonstrate the great potential to improve solubility, cellular specifically targeted to intracellular sites of action, transcytosis across the tight epithelial and endothelial barriers, as well as the combination of therapeutic modalities with multifunctional drugs and therapeutics with imaging agents to visualize drug delivery and drug efficacy in real time [13]. In addition, the biodiversity of therapy and pharmacokinetics could be improved through EPR effect of different nanomaterials. As is known, PEG has been applied widely due to good biocompatibility, water solubility, and low immunogenicity. Moreover, PEG can increase circulation, reduce immunogenicity, and regulate drug pharmacokinetics, making it a perfect polymer backbone for drug coupling. HPMA exhibits similar properties to PEG to be explored as a hydrophilic polymer for therapeutic drug delivery [19]. Compared with PEG, HPMA has advantages in that HPMA copolymers do not trigger the production of anti-polymer antibodies [20,21,22]. To date, HPMA has been studied extensively for good biocompatibility, low immunogenicity, and broad choices.

Polymer nanomedicines are macromolecule-based, water-soluble, particular or micellar constructs that can be conjugated with carriers for targeted delivery and controlled release of biologically active molecules at specific sites [23]. Currently, HPMA copolymers have been used for the delivery of various drugs, such as in the fields of anti-inflammatory [24], antibiotic [25], and anticancer [26], to significantly reduce the overall toxicity of the loaded chemotherapeutic drugs, increase accumulation in inflammation or solid tumors, as well as enhance drug solubility, stability, and pharmacokinetics. The versatility of HPMA copolymers also provides a greater scope of selection for the binding of drugs and residues to facilitate the establishment of a comprehensive platform for combination therapy.

### 2.1. Tunable Chemical Structure

Other than the unique and excellent properties of HPMA, as mentioned above, HPMA is studied as a promising drug carrier due to its repetitive structure and N-substituted amide bonds. These chemical properties enable the addition of multiple targeting factors, therapeutic drugs, and tracer molecules as the building block of the HPMA chain [27]. Various functional groups can be introduced into the HPMA copolymer chain through different monomers; as a comparison, PEG does not exhibit this capability. The structure of PEG is more rigid, with the molecular weight of the long or short chain varying. Therefore, HPMA copolymers demonstrate favorable properties concerning selective choices.

A distinctive advantage of HPMA copolymers is the versatility of the polymer intermediates; the functional groups at the terminus of HPMA facilitate the coupling to the residues of therapeutic drugs and oligopeptides, which permits higher drug-carrying capacity. For example, monomers containing hydrophobic or hydrophilic functional groups such as methyl methacrylate (MMA) [28,29], glycidyl methacrylate (GMA) [30,31], 3-(3-methacrylamidopropanoyl) thiazolidine-2-thione (MA-AP-TT) [32,33], N-(3-aminopropyl) methacrylamide (APMA) [34,35,36,37,38,39], 2-hydroxyethyl methacrylate (HEMA) [40], N-methacryloylglycylglycine (MAGG) [41,42], acrylic acid (AA) [42], and N-(3-azidopropyl)methacrylamide (AzMA) [43,44] could be prepared with HPMA as HPMA copolymers (Figure 1). These monomers carrying different functional groups provide comprehensive options for conjugation.

In addition, the HPMA copolymers synthesized with different functional groups usually exhibit different physiochemical properties. The introduction of hydrophilic functional monomers as blocks can enhance the water solubility of the copolymers. Therefore, the drug delivery by this hydrophilic HPMA system can be easily dispersed and transported in an aqueous environment to improve the efficacy and bioavailability of the loaded drugs. Meanwhile, hydrophilic monomers could increase the biocompatibility of the copolymers, reducing irritation and adverse reactions to living organisms. In other cases, the introduction of hydrophobic monomers could make copolymers amphiphilic, allowing HPMA copolymers to be self-assembled in water to form micelles [45,46] and nanoparticles [47] (Figure 2), in which hydrophobic drugs could be encapsulated in the internal hydrophobic core with enhanced efficacy for delivery. Kramer et al., 2019 [48], developed amphiphilic block copolymers of HPMA using a dual orthogonal connection strategy. The developed copolymers can be further coupled with antibodies and radiolabels and can produce nanoparticles of various sizes. By modifying the composition of copolymers, copolymer precursors with different numbers of reactive groups randomly distributed along the copolymer chain could be synthesized, and different structural units could be linked to drugs, peptides, proteins, antibodies, dyes, or radionuclides, which broadens the applications of copolymer–drug couplings. The versatility of HPMA copolymers can be achieved through simple synthesis, reversible addition-fragmentation chain transfer (RAFT) [45,49], atom transfer radical polymerization (ATRP) [46], and click chemistry methods to generate the shape of the linear, diblock, star [50], or vesicular [51] copolymers.

### 2.2. Molecular Weight

As known, the accumulation of the high molecular weight of polymers in solid tumors is due to the EPR effect [52]. Low molecular weight drugs often lack tumor selectivity and lead to dose-dependent toxicity, which can be addressed by developing high-molecular-weight polymer–drug conjugates to enhance drug selectivity and reduce systemic toxicity. For tumor targeting, the molecular weight of the polymers should be sufficient to exceed the threshold of glomerular filtration in the kidney, which prevents the rapid elimination of drugs from living organisms and prolongs the circulation of polymers to realize maximal accumulation in targets. Thus, the efficiency of tumor accumulation of HPMA copolymers is molecular weight dependent [53] and would enhance with increasing molecular weight compared with the behavior of other polymers. For example, Noguchi Y prepared ^125^I-labelled HPMA copolymers and demonstrated a significant increase in tumor accumulation by the higher molecular weight copolymers (>40 kDa) 6 h after intravenous injection. Furthermore, the ratios of the tumor to normal tissue indicated strong molecular weight-dependent accumulation. In contrast, lower molecular weight copolymers (<40 kDa) were rapidly cleared from tumor tissue due to rapid diffusion back into the bloodstream, suggesting that HPMA copolymers greater than 40 kDa could lead to a higher uptake by the EPR effect in tumors [54]. It has been well established that the design of blocks [55,56], the length of chain [17,57], and the shape of structures significantly affect copolymers’ distribution and biological activity.

The molecular weight of HPMA copolymer has a significant effect on the performance of the drug delivery system. A larger molecular weight of the copolymer can increase the drug loading, prolong the circulation time, and improve the bioavailability of the drug. Using an initiator to carry out traditional free radical polymerization in solution to initiate chain polymerization of HPMA monomers is a relatively simple synthesis method. Still, its molecular weight distribution is wide (PDI > 1.5), and the controllability is poor, making it difficult to precisely control the chain length of the polymer [58]. HPMA copolymers are subjected to RAFT with the corresponding monomers. By changing the composition, copolymer precursors with different numbers of reactive groups randomly distributed on the copolymer chain are synthesized. This can achieve controllable free radical polymerization and design a narrow synthetic molecular weight distribution (PDI < 1.2), which can be used to accurately design block or graft copolymers [59,60] and dominate the design of targeted therapy and smart response carriers. However, Armin Azadkhah Shalmani and Zaheer Ahmed [61] compared the copolymer micelles prepared by the two methods, which showed similar size, size distribution, cell compatibility, and drug loading and retention capacity. Contrary to the expectation to synthesize polymers with narrow molecular weight distribution, low dispersity does not necessarily translate into micellar self-assembly with better drug properties. However, due to the copolymer’s too-large molecular weight, a decrease in water solubility would be observed, affecting the release of the conjugated drugs and the toxic accumulation of the copolymer in non-target tissues. Therefore, the molecular weight of HPMA copolymers needs to be precisely determined according to the specific drug’s therapeutic need. Commonly, the molecular weight of HPMA copolymers could be regulated by selecting different polymerization methods and reaction conditions. To generate a better drug delivery system, the molecular weight and distribution of blocks in the copolymers could be optimized by controlling factors such as the type of initiator, amount, reaction temperature, and time.

### 2.3. Solubility

Other than the site-specific targeting, drug solubility is also considered as the key for drug development. For example, the macrophage-targeting peptide M2pep selectively binds to M2-type macrophages, and the rigid peptide scaffold of tetravalent M2pep mediates M2 toxicity. However, this peptide is less water soluble for in vivo applications. To solve this problem, Chayanon Ngambenjawong et al., 2017 [62], attached three M2pep peptide analogs to HPMA copolymers to form M2pep grafted copolymers, which were more water soluble and serum stable with selective toxicity to M2-type macrophages.

Approximately, 40% of marketed drugs and 70% of active pharmaceutical ingredients (APIs) have poor aqueous solubility [63]. There are many hydrophilic groups, including hydroxyl groups, in the structure of HPMA copolymers. These hydrophilic groups form hydrogen bonds with water molecules and have good water solubility. Therefore, various strategies and techniques are being developed to enhance their solubility. Some small molecules, such as Paclitaxel (PTX), a highly hydrophobic antineoplastic drug with very low solubility in water, have been extensively studied for improving solubility [64]. It was shown that significant improvement of solubility can be achieved by combination with HPMA to form a polymeric prodrug, which is more conducive to in vivo drug delivery and absorption [59,65]. Poorly soluble drugs can be conjugated with HPMA to form copolymers, where the hydrophilic nature of HPMA enhances drug dispersion in the aqueous phase and significantly improves solubility. Triterpenoids are one of the most important classes of natural compounds in many plant species, and their derivatives have been studied for years. As the active components in the extraction of this class, Betulinol and betulinic acid (BA) exhibit a wide range of biological activities, including antiretroviral, antibacterial, anti-inflammatory, anticarcinogenic, antioxidant, and anthelmintic [66]. However, BA and its metabolic precursor, betulin, are virtually insoluble in water, which restricts their pharmaceutical applications. Ekaterina et al., 2016 [67], synthesized several compounds by conjugating HPMA and BA derivatives for passive tumor accumulation and effective delivery to solid tumors. The copolymers exhibited high cytostatic activity against DLD-1, HT-29, and HeLa cancer cell lines and enhanced tumor accumulation in HT-29 xenograft mice. Researchers have explored the use of HPMA as a key component in polymer micelles. HPMA can form the hydrophilic outer corona of the micelle, while, after chemical modification, its hydrophobic segments can serve as the micelle core, enabling the encapsulation and solubilization of hydrophobic drugs [68].

### 2.4. Biocompatibility

Biocompatibility is defined as the ability of a material to perform its intended function in medical treatment, to interact with living systems, to induce appropriate host responses in a specific application without any risk of injury, toxicity, or rejection by the immune system, and to avoid adverse or inappropriate local or systemic effects. It can be used to assess the interaction between materials and organisms to ensure that the materials do not cause harmful reactions in the body [69,70]. Studies have shown that the amount of polymers would decrease in some organs, such as the spleen and lymph nodes, initially with a climbing increase [71,72]. The following measurement has shown that the peak accumulation of polymers in the organism is especially located in the reticuloendothelial system (RES). Usually, this accumulation is temporary; with the gradual release of polymers from RES, the trapping is liberated without inducing any significant pathological changes. The accumulation of polymers with higher molecular weights can be more pronounced in phagocytosis-active organs without any changes by histological examination [71]. In the investigation of enhanced cellular uptake by copolymer delivery, the cytotoxicity of various blank copolymer carriers, including HPMA and APMA, was evaluated. No cytotoxicity was observed for conjugates, demonstrating the safety of copolymers in cells, except for a slight toxicity at the highest concentration of the positively charged copolymers [39].

After applying self-assembled HPMA copolymer as therapeutic and diagnostic nanomedicine, no red color in red blood cell (RBC) supernatant was observed at high concentrations. The percentage of hemolysis at maximum nanoparticle concentration was about 3.5%, which is lower than the acceptable percentage of hemolysis (5%), presenting good hematological compatibility of HPMA copolymer. In addition, the treated mice did not show any abnormal symptoms or behaviors throughout the experiments, and the H&E staining of organs did not indicate any pathological changes, confirming no potential toxicity of HPMA copolymer to normal tissues and organs [73].

## 3. HPMA Copolymers as Drug Delivery System for Peptides

As is known, the development of targeted drug delivery systems promises excellent outcomes for reducing the side effects of and improving the efficacy of chemotherapy by providing specificity to target tumor tissues or cells [74]. As a water-soluble polymer material, HPMA copolymer plays a crucial role in enhancing the physicochemical properties and biological performance of peptides, which are inherently prone to enzymatic degradation and aggregation within the body. However, conjugation with HPMA copolymers offers significant protection by shielding peptides from enzymatic breakdown and aggregation while simultaneously reducing immune system recognition. Furthermore, the hydrophilic groups of HPMA copolymers, such as hydroxyl groups, contribute to increased peptide hydration, further improving their stability and functionality. Thus, HPMA copolymers are used as carriers to deliver peptides to specific tissues or cells. The problem of HPMA copolymers is the lack of actively targeting moieties. As functional molecules, peptides can give HPMA copolymers new properties and functions after binding to HPMA copolymers. Peptides themselves may have biological activities (such as antibacterial and anti-tumor) [75], and their activity can be enhanced after binding to HPMA copolymers. By introducing specific ligands or antibodies on HPMA copolymers, couplings can be designed to selectively recognize and bind to receptors or antigens expressed on the surface of target cells or tissues [16,76]. When the HPMA copolymer–peptide circulated the tumor tissue, the targeting ligand could recognize the target cells and facilitate translocation into the cells (Figure 3). To optimize the precision, the ligand needs to present as highly specific for its cognate receptor, and the presence of HPMA copolymers enhances drug accumulation at the desired site and enhances the therapeutic effect while minimizing systemic side effects. The introduction of peptides can expand the functionality of the HPMA copolymer (such as enzyme responsiveness) [56]. The dynamic interplay between HPMA copolymers and peptides endows HPMA copolymer–peptide conjugates with immense potential in drug delivery systems, particularly in cancer therapy. In the following sections, the applications of HPMA copolymer–peptide through two key approaches, active targeted drug delivery systems (Table 1) and stimuli–responsive drug delivery systems, are summarized.

### 3.1. Active Targeted Drug Delivery System

#### 3.1.1. Bombesin

Bombesin (BBN) is an active tetra decapeptide initially isolated from the skin of two European amphibians, *Bombina bombina* and *Bombina variegata*, of the family Discoglossidae [77]. It is composed of fourteen amino acids with high cell permeability and biocompatibility. The binding receptors of BBN could be found in subtypes of interneuron B receptor (NMBR/BB1), gastrin-releasing peptide receptor (GRPR/BB2), and orphan receptor subtype (BB3). GRPR has been shown to be expressed in most human prostate cancers; therefore, diagnostic and therapeutic agents targeting GRPR-overexpressed cancers have been widely investigated [78,79]. On the other hand, researchers have been exploring the potential application of BBN peptides to enhance delivery efficacy in the design of nanomaterials for targeting cancer cells [80,81,82,83]. Sameer Alshehri et al., 2020 [36], prepared a series of innovative BBN peptide-conjugated HPMA copolymers with different charges by RAFT and labeled them with fluorescent dye FITC and radioactive isotope ^177^Lu. The positive charge of the HPMA copolymers could play a profound role in promoting surface binding during cellular uptake due to the good electrostatic interaction with the negatively charged plasma membrane. The cellular internalization of positively charged HPMA copolymers increased with the increasing BBN peptides and net charge ratio. However, the biodistribution profiles of the positive copolymers in normal mice indicated cleavable problems with rapid recognition and clearance by mononuclear phagocyte system-associated tissues [84,85]. Small truncated BBN peptides were conjugated to HPMA copolymers (ranging from 42 kDa to 101 kDa) to enhance the uptake of GRPR-positive tumors. The charge and incorporation density of the targeting carrier play crucial roles in determining the in vitro and in vivo performance [86]. Polymers with different charges were prepared to modify BBN, with HPMA copolymers increasing the amount of BBN, thereby enabling enhanced anti-tumor effects. Given that the overexpression of GRPR in cancer cells facilitates the selective targeting of cytotoxic drugs, HPMA copolymers effectively minimize BBN consumption during delivery, thereby enhancing the tumor-targeting potential.

#### 3.1.2. Cell-Penetrating Peptides

Cell-penetrating peptides (CPPs) are a class of short peptides with unique structures and functions. They are usually composed of no more than 30 amino acids. They can carry a variety of biologically active molecules, such as proteins, nucleic acids, and small molecules, across the cell membrane to penetrate the cytoplasm [87]. CPPs exhibit the advantages of high biosafety, low cellular toxicity, and flexible design. Based on the different properties of cell-penetrating peptides in terms of source, nature, and function, they can be classified into five categories, including cationic peptides, amphipathic peptides, hydrophobic peptides, proline-rich peptides, antimicrobial peptides, and chimeric peptides [88]. Currently, CPPs have shown great potential for application and development in various fields, such as drug delivery, bioimaging, and biomedical research [89,90].

In a related study, Shuhua Li et al., 2014 [91], synthesized HPMA copolymer coated with adenovirus-coupled activatable cell-penetrating peptide (ACPP), which acts as a polycationic peptide. Originally, ACPP is neutralized by polyanionic sequences through fusion of cleavable junctions and is only released as a polycationic peptide in the peripheral environment of the tumor area in the presence of matrix metalloproteinases (MMPs). This design enables the carrying substance to bind the cell and enter the cytoplasm of MMP-overexpressing tumor cells. Notably, the transport of these HPMA copolymers into the cytoplasm by ACPP was characterized by non-endocytosis and non-concentration dependence. In contrast, HPMA copolymer-coated Ad5 for adenovirus receptors without ACPP enters the cell only by endocytosis. Non-endocytic and non-concentration-dependent transport may circumvent lysosomal degradation and improve gene delivery efficiently. However, MMPs are highly expressed in various physiological processes, which may lead to off-target accumulation. The heterogeneity of MMP activity in the tumor microenvironment may limit universality. Yucheng Xiang et al., 2018 [92], connected human cell-penetrating peptide dNP2 with HPMA copolymer DOX conjugate to establish a tumor-targeted drug delivery system. The integrated system contains 5.78 wt% DOX drugs with a molecular weight of 27.62 kDa, lower than the glomerular filtration limit to ensure safe excretion through urine. Compared with the unmodified copolymer (P-DOX), the integrated system demonstrates good biocompatibility and stronger DNA damage ability at the cellular level, as well as inducing cell apoptosis and promoting more effective accumulation of anti-cancer drugs in tumor cell nuclei, thereby significantly improving the anti-cancer efficacy.

Antimicrobial peptides (AMPs) are cationic, amphiphilic molecules characterized by both hydrophobic and polycationic domains. Due to their ability to efficiently translocate across membranes, AMPs have been explored as drug delivery carriers capable of entering cells without compromising membrane integrity. Notably, they can penetrate the cell membrane at remarkably low concentrations (micromolar) in the absence of specific receptors. They can facilitate the intracellular delivery of bioactive drugs through electrostatic or covalent interactions [93]. Given their strong affinity for cancer cell membranes and ability to translocate into the cytoplasm while exhibiting lower affinity for normal cells, AMPs have emerged as promising candidates for enhancing the cellular uptake of peptide antigens and chemotherapeutic agents [94]. A notable example is a peptidomimetic derived from the antimicrobial peptide SVS-1, conjugated to an HPMA copolymer backbone with a molecular weight range of 19.9–22.1 kDa and synthesized via RAFT. This copolymer efficiently penetrates the cell membrane and rapidly translocates into the nucleus without compromising cell viability. Furthermore, when the chemotherapeutic agent DOX was conjugated to this SVS-1-modified HPMA copolymer (SVS-1-P-DOX), the penetrating capability of SVS-1 facilitated enhanced DNA damage, increased apoptosis, and led to superior cytotoxicity [37]. CPPs and drugs are encapsulated within water-soluble HPMA copolymers, altering their distribution profile within the body. Upon release in targeted regions, these conjugates efficiently penetrate cells and exert therapeutic effects. These studies further highlight the key value and great potential of cell-penetrating peptides in tumor-targeted therapy and provide a strong theoretical basis and practical reference for future clinical application of tumor treatment.

#### 3.1.3. Mitochondria-Targeted Peptides

Mitochondria play a critical role in cellular biology. On one hand, they serve as the “powerhouse” of the cell, generating adenosine triphosphate (ATP) through oxidative phosphorylation to generate energy for various physiological activities of the cell [95,96]. On the other hand, they act as a central hub for programmed cell death pathways, regulating apoptosis through key pro- and anti-apoptotic proteins such as Bax, Bak, and Bcl-2 [97]. Mitochondrial dysfunction has been implicated in numerous diseases, with cancer being a primary area of concern. The occurrence, development, and metastasis of many tumors involves alterations in mitochondrial function. Beyond cancer, mitochondrial dysfunction is closely related to diseases such as diabetes and obesity as well as neurodegenerative diseases like Parkinson’s and Alzheimer’s [98]. Therefore, mitochondria-targeted therapy holds great promise not only for cancer treatment but also for treating other mitochondria-related diseases.

The mitochondrial-targeted drug delivery system is constructed by integrating a mitochondrial-penetrating peptide, a linear HPMA copolymer synthesized via free radical polymerization, and an anticancer drug linked through a pH-responsive hydrazone bond. This design enables precise drug delivery to mitochondria, enhancing therapeutic efficacy while minimizing off-target effects. SS20 peptide is a mitochondria-targeting peptide that selectively targets cardiolipin in the inner mitochondrial membrane without depending on mitochondrial membrane potential, which is altered in disease states [99]. Modification of SS20 with an HPMA copolymer (P-FITC-SS20) significantly altered its uptake profile, making its internalization independent of endocytosis inhibitors or temperature. Additionally, this modification avoids trapping by lysosomes and extends blood circulation time to enhance tumor accumulation [100]. Yanxi Liu et al., 2018 [101], further developed this strategy by incorporating α-tocopheryl succinate (α-TOS), a vitamin E derivative known to induce apoptosis in various cancer cells through rapid production of reactive oxygen species (ROS). To enhance mitochondrial targeting, they designed and synthesized an HPMA copolymer (P-TOS-SS20-dNP2) with a molecular weight of 35.3 kDa. The researchers effectively enhanced mitochondrial targeting by co-modifying HPMA copolymers with SS20 and dNP2. While the copolymer modified with SS20 alone exhibited limited mitochondrial localization and dNP2 alone failed to improve mitochondrial accumulation, their combination successfully compensated for the deficiencies of each peptide, leading to a significant increase in internalization and enhanced mitochondrial accumulation. The modified copolymer (P-TOS-SS20-dNP2) demonstrated potent anti-proliferative effects, tumor spheroid growth inhibition, and increased apoptosis and necrosis, validating SS20-based modification as a viable strategy for mitochondria-targeted drug delivery and can be further applied to mitochondria-associated diseases to improve therapeutic efficacy. Meanwhile, Qiuyi Li et al., 2020 [102], also explored multi-peptide strategies for mitochondrial targeting in tumor cells. They synthesized HPMA copolymers by simple free radical polymerization. They used them to combine the cell-penetrating peptide octopine (R8) and the mitochondrial targeting sequence ALD5MTS with DOX to develop a mitochondrial-targeted drug delivery system (P-D-R8MTS) aimed at synchronously inhibiting the progression and metastasis of breast cancer. Upon entering tumor cells, DOX-R8MTS was released in a pH-responsive manner in acidic lysosomes, enabling precise mitochondrial targeting. This method significantly enhanced the production of ROS and the initiation of cell apoptosis and effectively inhibited the proliferation, migration, and invasion of breast cancer cells by destroying mitochondria in vitro. These findings support the potential of mitochondria-targeted drug delivery for preventing and combating metastatic cancer.

Mitochondrial membrane-penetrating peptides (MPPs) offer an advantage over conventional CPPs due to their higher affinity for the hydrophobic mitochondrial membrane, facilitated by introducing hydrophobic amino acids [103,104]. In addition to promoting cellular uptake as CPPs, MPPs can realize mitochondrial targeting, making them particularly promising against multidrug resistance (MDR)—a major challenge in chemotherapy [105]. Mitochondria-targeted HPMA copolymers modified with MPPs significantly enhance drug accumulation within mitochondria, leading to increased ROS production and ATP depletion (Figure 4). This process, in turn, disrupts drug efflux mechanisms, allowing for more effective drug retention within resistant cancer cells. For instance, HPMA copolymers were engineered to bypass drug efflux pumps through endocytosis, successfully delivering DOX to drug-resistant MCF-7/ADR breast cancer cells, thereby inhibiting tumor growth [106]. Further optimization of this strategy led to the development of MPP-modified, DOX-loaded HPMA copolymer conjugates (PMNs) designed to target both mitochondria and the nucleus. The PM component facilitated direct mitochondrial delivery, countering metastasis by impairing mitochondrial function, while the PN component capitalized on DOX’s nuclear accumulation to induce DNA damage. The co-delivery of these two conjugates exhibited complementary effects, with PM suppressing metastasis and PN inhibiting tumor growth. Additionally, their synergistic action induced apoptosis via distinct pathways, resulting in significantly enhancing apoptosis initiation and significant damage to tumor cells [107]. Given the critical role of mitochondria in drug resistance and cancer progression, further in-depth research into the properties and functions of mitochondria-targeting peptides is essential for advancing cancer therapy and treating other mitochondria-related diseases. The development of more efficient and specific mitochondria-targeting peptides for drug delivery systems might focus on designing more efficient and specific mitochondria-targeting peptides to improve drug delivery precision, optimizing peptide combinations and HPMA copolymer structures to enhance targeting, therapeutic efficacy, and safety, expanding research on bi-peptide and multi-peptide modifications across different tumor types and disease models; bridging the gap between laboratory research and clinical application by conducting in-depth studies on pharmacokinetics, biocompatibility, and long-term safety in vivo. By refining peptide-based modifications and optimizing delivery strategies, mitochondria-targeted therapy holds immense potential for cancer treatment and anti-MDR, paving the way for more effective and clinically viable therapeutic interventions.

#### 3.1.4. αVβ_3_ Targeting Peptides

The integrin α_V_β_3_ receptor is a highly attractive target in cancer therapy, as it plays a pivotal role in tumor angiogenesis by regulating endothelial cell adhesion, migration, proliferation, and apoptosis [108,109]. Notably, α_V_β_3_ is overexpressed on the surface of various malignant tumor cells and tumor-associated endothelial cells. In contrast its expression in normal tissues remains relatively low [110], making it an ideal target for selective drug delivery and tumor imaging.

The transmembrane glycoprotein FQSIYPpIK (FQS) peptide exhibits a high affinity for the integrin α_V_β_3_ receptor [111]. Studies have shown that the conjugation of HPMA with FQS effectively facilitates α_V_β_3_-targeted drug delivery, demonstrating the critical role of FQS in targeting [112]. Another well-established targeting motif is the Arg-Gly-Asp (RGD) tripeptide, which exhibits a strong binding affinity for α_V_β_3_. The exogenous RGD peptide competitively binds to integrin receptor α_V_β_3_, allowing for the detection of α_V_β_3_ expression and inhibition of tumor cardiovascular angiogenesis, ultimately leading to tumor necrosis or apoptosis.

Daniel B. Pike et al., 2010 [113], utilized HPMA copolymer-cyclic RGD conjugates to target tumor angiogenesis selectively. Their study showed that HPMA–RGD4C and HPMA–RGDfK conjugates exhibited increased accumulation in prostate, lung, and breast tumors compared to the free peptide. RGD4C did not affect its activity and affinity for αVβ_3_ when conjugated to HPMA. Targeted accumulation was also found after labeling by ^99m^Tc or ^90^Y for imaging or therapy. While RGD-conjugated HPMA copolymers enhance drug delivery efficiency, the nonspecific accumulation of therapeutic radionuclides in non-target organs—particularly in the liver, spleen, and kidneys—remains a challenge, which can lead to severe radiotoxicity and limit the clinical applicability of these conjugates.

Based on this strategy, Zheng-Hong Peng and Jindřich Kopeček et al., 2015 [114], developed a matrix metalloproteinase-2 (MMP-2)-responsive HPMA copolymer-iRGD-DOX conjugate (MA-GG-PLGLAG-iRGD) by incorporating an MMP-2-cleavable PLGLAG spacer. The introduction of iRGD into the drug carrier system significantly enhanced DOX penetration and accumulation in both monolayer and multicellular spheroid models of prostate cancer. This improvement facilitated cell cycle arrest and apoptosis, demonstrating the efficacy of iRGD-mediated drug delivery in enhancing penetration and therapeutic outcomes. However, since this effect of the conjugate was observed only in vitro, further in vivo studies are needed to evaluate its therapeutic efficacy and potential toxicity in normal tissues.

RGD peptides offer precise tumor-targeting capabilities, making them a promising tool for integrin α_V_β_3_-mediated drug delivery. By selectively binding to integrin αVβ_3_ on the surface of tumor cells and angiogenic endothelial cells, RGD can directly deliver drugs or other therapeutic molecules to the tumor cells to enhance treatment efficacy and reduce off-target effects for targeted treatment. After RGD is connected to an HPMA copolymer, the aggregation of RGD in the tumor site is increased, the uptake of non-targeted organs is reduced, the therapeutic effect of the anti-tumor drugs carried on the copolymer is enhanced, and the adverse reactions are effectively reduced. Significantly, the conjugation of RGD with HPMA copolymers does not compromise its binding affinity to α_V_β_3_ or biological activity, which means the targeting capability and inhibition to tumors remain to maximize the therapeutic efficacy. Furthermore, the RGD–HPMA copolymer model provides a versatile platform for developing multimodal drug delivery systems. Integrating chemotherapeutic agents, gene therapy vectors, or immunotherapeutic molecules with RGD–HPMA copolymers can provide personalized therapeutic regimens for different types of tumors.

#### 3.1.5. Nucleus-Targeting Peptide (NrTP6)

The nucleus is a central regulator of cell metabolism, growth, and differentiation, making it an attractive target for various disease treatments, particularly cancer. The c-Myc protein, a nuclear transcription factor, is strongly associated with malignant tumor phenotypes, regulating cell proliferation, gene transcription, and tumor metabolism [115]. Studies have shown that the H1-S6A and F8A (H1) peptides can specifically bind to c-Myc, effectively inhibiting its function and consequently suppressing tumor cell proliferation to achieve therapeutic effects [116].

The HPMA copolymer conjugated with nucleolus-targeting peptide (NrTP6) has been developed to improve nuclear-targeted peptide therapy. This platform significantly enhances the nuclear delivery of H1-S6A and F8A peptides and prevents c-Myc from binding to DNA to play an oncogenic role. Compared to non-NrTP6-modified H1 peptide conjugates, NrTP6-modified HPMA-H1 peptide conjugates exhibit a 2.2-fold increase in cellular internalization and a 37.1-fold increase in nuclear accumulation [117]. NrTP6 modification represents a promising strategy for simultaneous cellular uptake and nuclear targeting, providing a novel approach for intranuclear drug delivery. This strategy could be further explored to enhance precision-targeted cancer therapies by disrupting nuclear oncogenic pathways, ultimately improving therapeutic efficacy and specificity in tumor treatment.

**Table 1 biomolecules-15-00596-t001:** HPMA copolymer delivery system with actively targeted peptide.

Peptides	Therapeutics	Imaging Probes	Cell Line	Types of Cancer	Outcomes	Refs
BBN	^177^Lu		PC-3	Prostate	The positively charged conjugates were more efficiently taken up by PC-3 cells than the negatively charged ones but were cleared from the blood within 4 h in normal CF-1 mice.	[36]
Bradykinin	/	/	C26	Colon	pH-responsive HPMA-BK copolymers increased blood flow to tumor tissue by 1.7-fold and selective accumulation in tumor tissue by 3-fold.	[118]
Collagen hybridizing peptides (CHPs)	/	Cy5	MDA-MB-231	Breast	HPMA copolymer-CHP exhibited a selective affinity for denatured collagen, and the conjugates increased tumor localization and showed higher retention compared to free CHP.	[41]
ACPP	Ad5	/	A549, MDA-MB-231, HepG2 and HBE	/	ACPP promotes pc-Ad-eGFP to directly cross the cell membrane and enter the cytoplasm, facilitating the efficient expression of therapeutic adenovirus.	[91]
dNP2	DOX	/	HeLa	Cervical	The DNA damage ability of P-(dNP2)-DOX is 10.5 times higher than that of P-DOX, and the inhibition rate of 3D tumor spheroids is 78%.	[92]
SVS-1	DOX	Cy5	HeLa	Cervical	SVS-1-P-DOX showed 2.4-fold increased cytotoxicity compared with P-DOX and exhibited prolonged blood circulation and preferential tumor accumulation compared with free DOX.	[37]
SS20	/	FITC, Cy5	HeLa	Cervical	Tumor accumulation of P-SS20-Cy5 was higher than that of P-Cy5 (approximately 1.69-fold) within 96 h after administration.	[100]
SS20 + dNP2	α-TOS	FITC	HeLa	Cervical	The combination of these two functional peptides resulted in a 7.6-fold increase in cellular uptake and accumulation in mitochondria, increasing apoptosis and necrosis.	[101]
R8 + MTS	DOX	/	4T1, MDA-MB-231	Breast	P-D-R8MTS exhibited the highest anti-tumor efficacy of 86.8% and tumor accumulation was 2.98-fold higher than that of free DOX.	[102]
MPP	DOX	/	4T1	Breast	pH-responsive destruction of cell nuclei and mitochondria exerts anti-tumor and anti-metastasis effects.	[107]
KLA	/	FITC, Cy5	B16F10	Melanoma	The cellular uptake and mitochondrial targeting abilities of the targeted copolymer increased by 4.3-fold and 23.8-fold, respectively; the in vivo tumor inhibition rate reached a peak of 82.9%.	[119]
FQS	DOX	/	B16F10	Melanoma	The antitumor efficiency of FQS-HPMA-DOX (83.9%) was significantly higher than that of HPMA-DOX (44.9%).	[112]
RGD	/	^111^In	LLC1	Lung	HPMA copolymers of mono-RGDfK and di-cyclized RGD4C both enhance tumor uptake and reduce background accumulation.	[120]
	Aminohexyl geldanamycin (AH-GDM)	/	PC-3	Prostate	The conjugate exhibited similar binding activity to the free peptide and was better tolerated in vivo than the free drug.	[121]
		/	DU-145	Prostate	The higher permeability of the RGDfK-targeted conjugate at a lower dose showed higher anti-tumor activity in nude mice.	[122]
	/	Indocyanine green azido derivative (ICG)	U87-MG LN-18	Glioblastoma	The targeted fluorescent nanoprobe showed a 35% increase in accumulation in glioblastoma compared to nontargeted controls.	[123]
NrTP6	H1-S6A, F8A (H1)	FITC	HeLa	Cervical	NrTP6-modified HPMA copolymer-H1 peptide conjugates could improve internalization and nuclear accumulation of H1 peptide by 2.2 and 37.1-fold; tumor growth inhibition rate of 77.0%.	[117]
Microtubule-dependent nuclear-targeting peptide (MT)	H1-S6A, F8A (H1)	FITC	HeLa	Cervical	The apoptotic cells induced by H1, P-H1, and P-H1-MT were 20.8%, 26.5%, and 45.4%; nuclear drug accumulation was 15.8-fold higher than that of polymers without peptides.	[124]
Nuclear-homing cell-penetrating peptide (R8NLS)	DOX	FITC, Cy5	HeLa	Cervical	It exhibited 4.5-fold higher nuclear accumulation than HPMA-Dox and inhibited tumor growth by 75%.	[125]

### 3.2. Peptide-Triggered Drug Release System

The tumor microenvironment is rich in enzymes that play a crucial role in tumor cell infiltration and metastasis. Their concentration and activity are significantly higher than those found in normal tissues. To leverage this characteristic, enzyme-sensitive polymer drug delivery systems have been developed by incorporating small peptides susceptible to enzymatic hydrolysis into the polymer structure. MMPs are a family of zinc-dependent proteases involved in the degradation and metabolism of the extracellular matrix (ECM). These enzymes are overexpressed in almost all types of human tumors, contributing to tumor growth, invasion, and metastasis [126,127]. To exploit this tumor-specific enzymatic activity, MMP-responsive peptides have been incorporated into nano-drug delivery systems for targeted gene and drug delivery [114,128]. Marzieh Najafi et al., 2020 [46], designed and developed a new type of injectable hydrogel, “HyMic”, consisting of CCL micelles crosslinked by enzyme-cleavable substrates of MMP2 and MMP9. Upon enzymatic degradation, HyMic is converted into CCL gel and absorbed by HeLa cells, demonstrating the targeting effect through a three-step cascade design of “enzyme-responsive conversion → micelle release → ligand-mediated endocytosis”.

Cathepsin B is a member of the cysteine protease family and plays an important role in tumor invasion and metastasis. As a tetrapeptide sequence reactive to cathepsin B, Gly-Phe-Leu-Gly (GFLG) can circulate stably in the blood and is finally cleaved by the cathepsin B that is highly secreted around tumor cells [129,130]. Based on this strategy, HPMA copolymer PK1 (FCE28068) was developed, in which the anticancer drug DOX is conjugated via a GFLG linker. Similarly, HPMA copolymer PK2 (FCE28069) was designed with DOX and the liver-targeting moiety galactosamine. Both formulations entered clinical trials (Table 2, Figure 5).

PK1 was the first polymer–drug conjugate to enter clinical trials, undergoing safety and pharmacokinetics evaluation in phase I studies, and the results showed that toxicity was controllable, with a maximum tolerated dose (MTD) of 320 mg/m^2^ (based on DOX). However, phase II trials, which tested a dose of 280 mg/m^2^ for non-small cell lung cancer, breast cancer, and colon cancer, yielded disappointing results, with only limited positive responses [131]. The low response rate may be attributed to variations in cathepsin B expression among patients, as the therapeutic efficacy and pharmacokinetics of PK1 are influenced by vascular characteristics and enzymatic activity. In contrast, PK2, which incorporates galactosamine for liver targeting, exhibited improved tumor specificity compared to PK1 in patients with liver cancer. However, as the disease progressed, the expression of sialic acid glycoprotein receptors in tumor cells declined relative to normal liver cells. This resulted in a lower accumulation of radiolabeled drugs in tumors (3.2%) compared to normal liver tissue (16.9%), ultimately leading to the termination of PK2 in phase II clinical trials [132]. Similarly, in PNU166945, a drug–polymer conjugate utilizing the GFLG linker to enhance solubility, drug loading remained relatively low. The phase I clinical study was prematurely terminated due to severe neurotoxicity, preventing the determination of a dose-limiting toxicity threshold [133].

**Table 2 biomolecules-15-00596-t002:** List of HPMA copolymer conjugates approved for clinical trials.

Conjugate	Drug	Spacer	Refs
PK1 (FCE28068)	DOX	GFLG	[131,134,135]
PK2 (FCE28069)	DOX (Galactosamine)	GFLG	[136,137,138]
PNU166945	PTX	GFLG	[65]
MAG–CPT (PNU 166148)	CPT	/	[139]
AP5280	Carbo platinate	GFLG	[140,141]
ProLindac^TM^ (AP-5346)	Oxaliplatin	/	[142]

These clinical findings highlight that while HPMA polymer-based drug delivery systems are designed to mitigate chemotherapy-related side effects and enhance efficacy, their therapeutic success is still hindered by tumor microenvironment heterogeneity, dynamic receptor expression changes, and the combined toxicity of both the carrier and the drug. In future, the therapeutic potential of HPMA platforms need to be optimized by the development of intelligent responsive carriers, multimodal targeting strategies, and precise patient stratification.

Based on the cathepsin B-reactive tetrapeptide sequence, Hao Cai et al., 2020 [73], integrated the anticancer drug PTX, the fluorescent dye cyanine 5.5 (Cy5.5), and the MRI contrast agent Gd-DOTA into a branched pHPMA polymer using RAFT polymerization, click chemistry, and chemical complexation. The resulting product self-assembles into therapeutic diagnostic nanodrugs (BP-PTX-Gd NPs) (Figure 6). Several key structural features contribute to the enhanced performance of these nanodrugs, including their high molecular weight, stable linker chemistry, slightly negative surface charge, and branched polymer architecture with excellent chain flexibility and deformability. These properties collectively lead to a significantly prolonged blood half-life. Once in the tumor microenvironment, where cathepsin B is overexpressed, the polymer backbone undergoes enzymatic degradation into small fragments. These fragments are then efficiently cleared from the body, preventing the toxic accumulation of both the delivery vehicle and the MRI contrast agent.

Similarly, Yang Yang et al., 2013 [47], incorporated a cathepsin B-sensitive peptide (GFLGKGLFG) into the main chain of a copolymer with hydrophilic and hydrophobic blocks to synthesize biodegradable amphiphilic triblock HPMA copolymer–DOX conjugate-based nanoparticles as enzyme-sensitive drug delivery carriers. Compared to free DOX, these polymeric nanoparticles exhibited significantly enhanced antitumor efficacy.

By incorporating enzyme-responsive peptide fragments into the polymer structure, these nanomedicines achieve precise, tumor-specific drug release. The branched polymer architecture not only extends circulation time but also enhances tumor penetration. Moreover, their integrated theranostic design allows for real-time treatment monitoring, paving the way for smart nanomedicines with targeted therapy and multimodal imaging capabilities. However, several challenges remain, including tumor microenvironment heterogeneity affecting enzyme responsiveness, limited penetration efficiency of polymer carriers, complex multi-step synthesis processes, potential long-term accumulation risks, and the absence of standardized clinical translation protocols. Future efforts should focus on optimizing the universality of enzyme responsiveness, developing controlled metabolic pathways, and establishing scalable production systems to accelerate clinical applications of these promising polymer–drug conjugates.

## 4. Conclusions and Outlook

HPMA copolymers have emerged as highly promising polymer nanocarriers due to their excellent biocompatibility and versatile structural designs, endowing them with a wide range of functionalities.

From the perspective of material structure, linear and grafted HPMA copolymers feature relatively simple architectures. By modifying their side chains, they can achieve higher drug-loading capacities, while the incorporation of active targeting ligands significantly enhances tumor cell recognition [114]. Additionally, HPMA copolymers can be combined with materials such as polylactic acid to form amphiphilic structures. Leveraging the self-assembly process, these copolymers enable the effective encapsulation of hydrophobic drugs, thereby improving drug solubility and optimizing controlled-release profiles [143]. Copolymer–drug conjugates incorporating targeting moieties, ligands, antibodies, and peptides have emerged as a promising approach for cancer treatment. The targeted drug delivery system of HPMA copolymer–peptide can be strategically designed by integrating functionalized peptides—such as targeting and cell-penetrating peptides—with biocompatible HPMA copolymers. This approach enables the construction of nanocarriers with tumor-targeting capabilities, microenvironment responsiveness, and enhanced cell penetration. Additionally, leveraging the modifiability of HPMA copolymers, pH-sensitive bonds or enzyme-responsive linkers can be introduced to enable conditional drug release at the tumor site. Drug loading can be achieved through either chemical conjugation or physical encapsulation, ensuring efficient delivery. However, current challenges, such as insufficient targeting accuracy and limited in vivo stability, necessitate further optimization. Strategies such as dual-targeting approaches, increasing HPMA cross-linking density, or employing humanized peptide designs can significantly improve therapeutic efficacy. The versatile side-chain functional groups of HPMA copolymers allow for flexible conjugation with drugs, ligands, imaging agents, and immunomodulatory molecules. Their high peptide-loading capacity, tunable hydrophilicity/hydrophobicity, and multifunctionality facilitate the precise delivery of anticancer drugs with high efficacy and minimal toxicity. These advancements lay a strong foundation for the intelligent design of next-generation drug delivery systems, integrating multimodal treatment strategies for improved therapeutic outcomes.

Over the years, various HPMA copolymer-based drug carriers have been designed, synthesized, and evaluated for therapeutic efficacy. While several HPMA copolymer-related drugs have demonstrated strong anti-cancer activity or imaging potential in preclinical animal models, only a few—such as Doxorubicin (FCE28068, FCE28069), Paclitaxel (PNU 166945), Camptothecin (PNU 166148), and Platinum complexes (AP5280, AP5346)—have been prompted to clinical trials. However, none of them have been approved for clinical use or commercialization to date.

The primary challenge hindering the clinical translation of HPMA copolymers extends beyond targeting accuracy or sustained-release efficiency. A critical barrier lies in the batch-to-batch variability in peptide modification and copolymer synthesis. Factors such as peptide spatial conformation and the distribution of polymerization degrees can significantly impact the reproducibility of these carriers. This lack of consistency ultimately impedes clinical breakthroughs, highlighting the need for more standardized and scalable manufacturing processes to ensure reliability and regulatory approval. Despite these challenges, the recent advances of new synthetic techniques such as RAFT polymerization, click chemistry, and orthogonal covalent binding, as well as the development of a variety of new spacer structures, are expected to accelerate the clinical translation of HPMA-based copolymer candidates. These next-generation HPMA-based drug conjugates have potent anti-tumor effects, effectively inhibiting cancer cell proliferation and tumor angiogenesis. The novel and reasonable design for tailored HPMA copolymers as a precision-targeted drug delivery system could realize organ- or cell-specific treatment. In future, clinical progress crucially relies on identifying novel molecular targets, optimizing targeting moieties, and selecting appropriate therapeutic agents; through intelligent response design, combination therapy, and precision medicine strategies combined with synthetic technology innovation, we will promote it from laboratory to clinical application. In addition, developments in imaging techniques have enabled advances in polymers for therapeutic and theragnostic applications; the advances in cell biology have opened up new possibilities for investigating specific targets to optimize the selection of suitable targeting moieties and bioactive compounds, as well as a deeper understanding of cellular and subcellular regulation for newly designed polymers. These innovations pave the way for next-generation polymer therapeutics with enhanced efficacy, specificity, and clinical applicability.

## Figures and Tables

**Figure 1 biomolecules-15-00596-f001:**
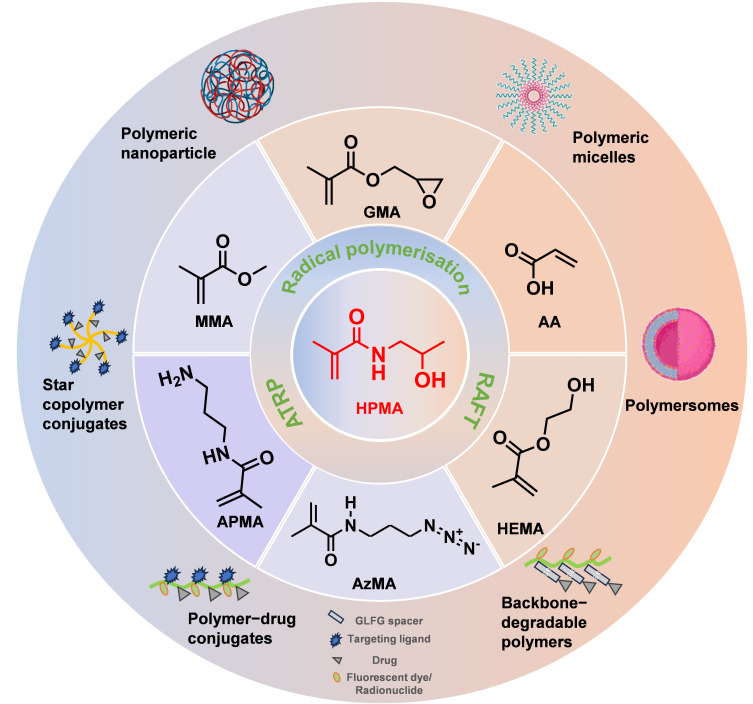
Nanostructures based on HPMA copolymers.

**Figure 2 biomolecules-15-00596-f002:**
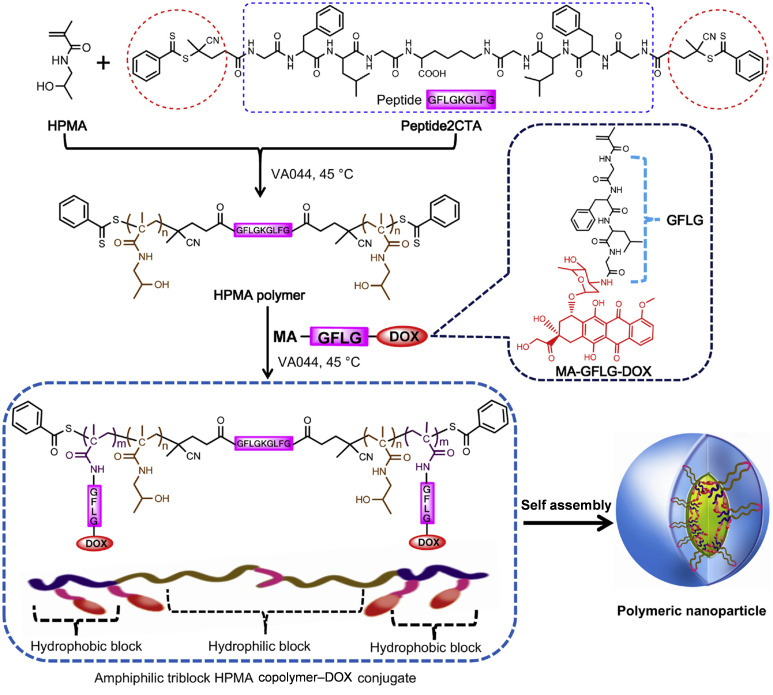
Structures and synthesis of amphiphilic triblock HPMA copolymer–doxorubicin (DOX) conjugate and the illustration of its polymeric nanoparticle. Reproduced with permission [47]. Copyright 2013, Elsevier.

**Figure 3 biomolecules-15-00596-f003:**
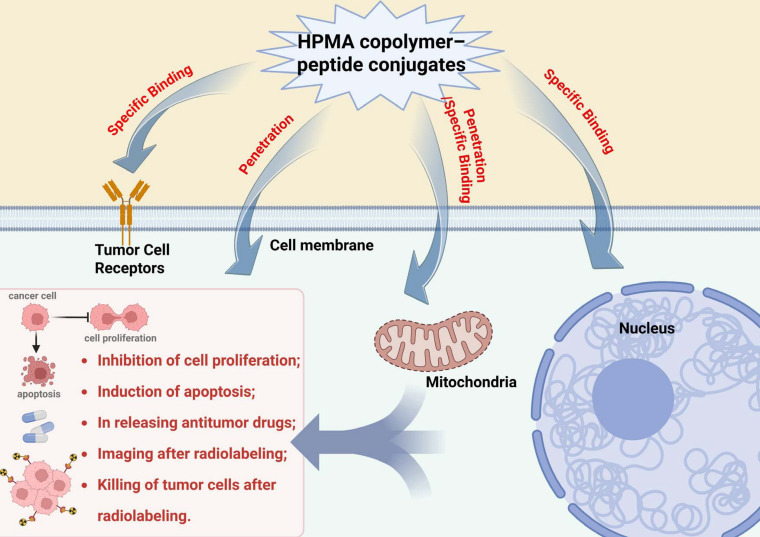
The role of HPMA copolymer–peptide drug delivery system. HPMA copolymer–peptide conjugates are connected to targeting peptides, anti-tumor drugs, radionuclides, and target tumor cell membranes and cell nuclei through the action of receptor ligands, direct penetration into the cell membrane for releasing drugs at specific locations, through the action of peptides to enter the cell mitochondria by enabling the drugs to inhibit cell proliferation and induce cell apoptosis, or the direct killing of tumor cells through the action of radioactive drugs.

**Figure 4 biomolecules-15-00596-f004:**
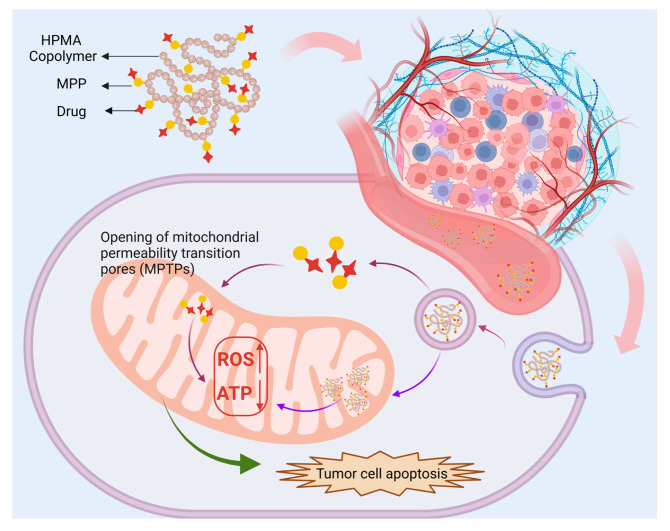
HPMA copolymer connects MPP with anti-tumor drugs. After entering tumor cells, drugs carrying MPP are released into mitochondria, or the targeted drug delivery system enters mitochondria to play a role. This induces the opening of the mitochondrial permeability transition pore (MPTP), which causes an imbalance of ROS/ATP and, eventually, cell apoptosis.

**Figure 5 biomolecules-15-00596-f005:**
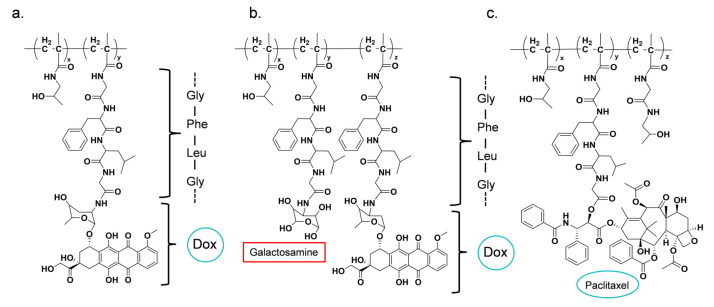
(**a**) PK1; (**b**) PK2; (**c**) PNU166945.

**Figure 6 biomolecules-15-00596-f006:**
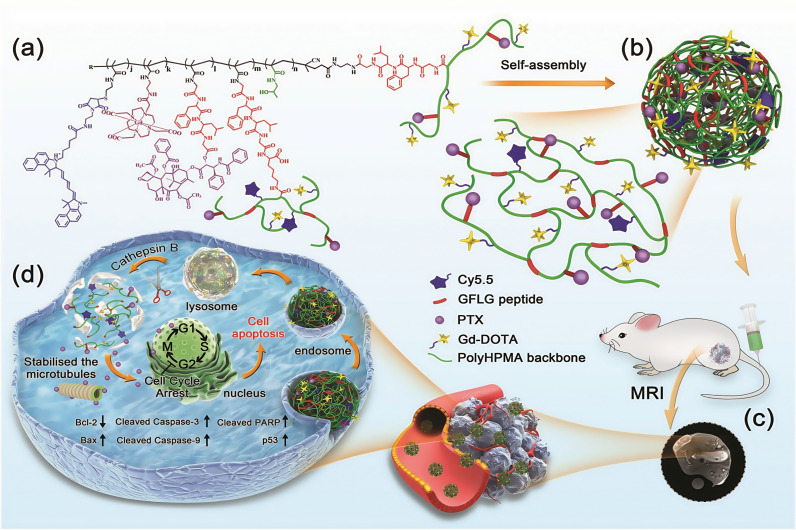
Schematic illustration of cathepsin B-responsive biodegradable theranostic nanomedicine derived from branched pHPMA. (**a**) Chemical structure of branched pHPMA-PTX-Gd conjugate; (**b**) Self-assembled BP-PTX-Gd nanoparticles; (**c**) In vivo magnetic resonance imaging of BP-PTX-Gd NPs delivered to tumor sites via the EPR effect; (**d**) After internalization via the lysosomal pathway, BP-PTX-Gd NPs degrades and releases PTX in a microenvironment with high expression of cathepsin B; the released PTX inhibits mitosis by stabilizing microtubules and activates the apoptotic pathway to induce cell apoptosis. Reproduced with permission [73]. Copyright 2020, WILEY.

## Data Availability

No new data were created or analyzed in this study.

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
