# Peer review of "HPMA Copolymers: A Versatile Platform for Targeted Peptide Drug Delivery"

_biomolecules, 2025, doi:10.3390/biom15040596_

Round 1
Reviewer 1 Report
Comments and Suggestions for Authors
The manuscript presents a review of HPMA-based polymers for applications in drug delivery. Although this is an interesting subject I cannot recommend the publication of the manuscript in the present state. Despite the use of almost 115 references, it is very difficult for the reader to follow all the examples provided and to conclude rational guidelines for all the diverse developments described by the authors. Second, the style of the manuscript is not quite consistent with the expectation of the readers of a journal that is focused on polymer science. The figures are exclusively bio-tutorial like. Reformatting is needed and more equations, structures and figures should be included. The discussion should focus on recent advancements of polymer science that have led to progress in the field, rather than just listing the materials.
- The title is not appropriate and does not reflect the scope of the manuscript.
- The abstract should be reformulated.
- The introduction should detail the difficulties of delivering peptides and why polymers are important for their protection. In addition, there are long paragraphs without references.
- There is a huge presentation of small summaries of papers read without any detailed and critical analysis of characteristics, results and efficacy.
- No comments about biological results.
- In principle, one could better classify copolymers into two classes: a) copolymers that are required to interact with the peptide (covalently, physically, by hydrophobic/ electrostatic/ supramolecular interactions etc, …) for the delivery and b) copolymers that are required to protect/ shield/ solubilize the drug. For the latter group, it is very important to also take into account that for parenteral applications these copolymers may also provide a "stealth" property in analogy like PEG - in fact, there are several reports vinyl-based monomers providing such properties (e.g. HPMA, etc.) and there are also new reports out on how they control the formation of a protein corona around nanocarriers decorated with these polymers. Some breakthrough examples in that field using HPMA copolymers are worth getting mentioned in this manuscript, as they describe also future development strategies for that field.
- HPMA-based micelles loaded with chemotherapeutics. They have even reached clinical trials (translated by the company Crystal Therapeutics). To the best of my knowledge, this is one of the few ongoing examples using vinyl polymers for parenteral applications as systemically administered polymer drug formulations - they are currently studied in the clinics and need therefore to be mentioned in the manuscript.
- No effort to describe what types of carriers might be more desirable.
- No discussion of methods of loading, preferred formulations, etc
- And why only very generic and unimportant figures? But in general, more Figures are needed.
- There should be more discussion of peptide-polymer conjugates.
- In sum, the authors should provide a detailed and comprehensive overview of the different types of HPMA-based peptide formulations that have been investigated in research as well as clinical trials or have even been translated into products on the market.
English should be improved and more careful.
Author Response
Thanks very much for reviewing this manuscript and for providing the opportunity to revise and resubmit the paper entitled “HPMA copolymers in the optimization of peptides” (biomolecules-3533683). We sincerely thank the editor and reviewers for their time and expertise in evaluating our work. All comments were carefully addressed to improve the quality and clarity of the manuscript. Please see the attachment for a point-by-point response.

Reviewer 2 Report
Comments and Suggestions for Authors
The title of this manuscript is “HPMA copolymers in the optimization of peptides”. From this title, the reader could think that the use of HPMA copolymers may help to optimize the preparation of these peptides. As the focus of this review is more related to the activity of peptides supported on HPMA copolymers, the title should be modified.
Then a Table containing a summary of the peptides described in the review together with the results obtained with peptides supported on HPMA copolymers should be added, to help the reader. The comment of this table should contain a critical comparison of the peptides supported on HPMA copolymers with the same peptides that are not supported, to show the effect of the HPTA support: is it effective? How much?
Author Response
Thank you very much for reviewing this manuscript and for providing the opportunity to revise and resubmit the paper entitled “HPMA copolymers in the optimization of peptides” (biomolecules-3533683). We sincerely thank the editor and reviewers for their time and expertise in evaluating our work. All comments were carefully addressed to improve the quality and clarity of the manuscript. Below is our point-by-point response to the reviewers’ suggestions, with revisions highlighted in red throughout the manuscript. Please see the attachment for the point-by-point response.

Round 2
Reviewer 1 Report
Comments and Suggestions for Authors
The quality of the article has been significantly improved. The authors have made all the requested suggestions/corrections. The article can be accepted for publication.